Effects of mixing eggs of different initial incubation time on the hatching pattern, chick embryonic development and post-hatch performance

Zhong Zhentao
Yu Yue
Jin Shufang
Pan Jinming panhouse@zju.edu.cn
Department of Biosystems Engineering, Zhejiang University , Hangzhou , China
Loor Juan
Electronic publication date: 2018 Apr 10
Publication date: 2018
Volume: 6
Electronic Location ID: e4634
Received 2018 Jan 17; Accepted 2018 Mar 28
Copyright: © 2018 Zhong et al.
Copyright year: 2018
Copyright holder: Zhong et al.
License: This is an open access article distributed under the terms of the Creative Commons Attribution License, which permits unrestricted use, distribution, reproduction and adaptation in any medium and for any purpose provided that it is properly attributed. For attribution, the original author(s), title, publication source (PeerJ) and either DOI or URL of the article must be cited.
License URL: https://creativecommons.org/licenses/by/4.0/

Keywords: Hatch sychrony, Egg redistribution, Hatch window, Embryonic development, Post-hatch performance

Funding: National Natural Science Foundation of China 31772644 This work was supported by the National Natural Science Foundation of China (Grant No. 31772644). The funders had no role in study design, data collection and analysis, decision to publish, or preparation of the manuscript.

==============================
Background

The hatch window that varies from 24 to 48 h is known to influence post-hatch performance of chicks. A narrow hatch window is needed for commercial poultry industry to acquire a high level of uniformity of chick quality. Hatching synchronization observed in avian species presents possibilities in altering hatch window in artificial incubation.

Methods

Layer eggs which were laid on the same day by a single breeder flock and stored for no more than two days started incubation 12 h apart to obtain developmental distinction. The eggs of different initial incubation time were mixed as rows adjacent to rows on day 12 of incubation. During the hatching period (since day 18), hatching time of individual eggs and hatch window were obtained by video recordings. Embryonic development (day 18 and 20) and post-hatch performance up to day 7 were measured.

Results

The manipulation of mixing eggs of different initial incubation time shortened the hatch window of late incubated eggs in the manipulated group by delaying the onset of hatching process, and improved the hatchability. Compared to the control groups, chick embryos or chicks in the egg redistribution group showed no significant difference in embryonic development and post-hatch performance up to day 7.

Discussion

We have demonstrated that eggs that were incubated with advanced eggs performed a narrow spread of hatch with higher hatchability, normal embryonic development as well as unaffected chick quality. This specific manipulation is applicable in industrial poultry production to shorten hatch window and improve the uniformity of chick quality.

Introduction

In artificial incubation, the inherent characteristics of eggs (e.g., parental age, egg weight and egg storage time) and incubation conditions (temperature and CO2 concentration during the hatching phase) play a crucial role in embryonic development that results in the spread of hatch (De Smit et al., 2006; Ipek & Sozcu, 2017; Maatjens et al., 2014; Nangsuay et al., 2016; Tona et al., 2003, 2007; Willemsen et al., 2010b). The spread of hatch is evaluated as the degree of hatching synchrony which essentially contributes to the uniformity of newly hatched chicks. In general, hatch window which defined as the time between early-hatching and late-hatching varies from 24 to 48 h (Careghi et al., 2005; Decuypere et al., 2001). Thus, early hatched chicks will be held in the incubators with deprivation of feed and water until the entire batch of chicks hatch, rather than removed immediately upon hatching. The variability of delayed time with access to feed and water depressed the uniformity of post-hatch performance of the hatched chicks, including organ development, immune system activation, digestive enzyme stimulation and relative growth post hatch (Tona et al., 2003; Willemsen et al., 2010a).

In nature, precocial avian species can achieve a narrow hatch window by acceleration (Holmberg, 1991; Vince, 1964) or retardation (Vince, 1968) of hatching. This adaptive advantage of hatch synchrony enables the offsprings to avoid being abandoned by the parent bird and exposed to predators. Moreover, intraclutch hatch synchronization was found in the lesser snow goose (Davies & Cooke, 1983), pheasants and mallard ducks (Persson & Andersson, 1999), by shortening or prolonging the incubation period. In addition to this effect of sibling contact, the hatch process could also be affected by mixing eggs of different embryo developmental trajectory (Tona et al., 2013). However, no effective manipulations during incubation have been performed to shorten hatch window in commercial poultry production.

Therefore, the aim of the present study was to achieve a narrowed hatch window through the manipulation of mixing eggs of different growth curves. In addition, potential effects on embryonic development and post hatch performance were studied. Hatching time of individual chicks, hatch window, hatchability, yolk residue and organ development, body weight and leg bone development were compared between the control and manipulated groups.

Materials and Methods

All procedures in this study were approved by the committee of Care and Use of Animals of Zhejiang University, Hangzhou, China.

Experimental design

Hatching eggs (n = 680; weight range from 53 to 57 g) were obtained from a Hyline breeder flock at 35–36 weeks of age (Shenhai Breeding, Shenyang, China) and all eggs were laid on the same day and stored for no longer than two days. The eggs were marked with numbers and divided into early incubation group (EI) and late incubation group (LI). LI started incubation 12 h later than EI so that the biological age (BA, calculated from the initial incubation time) of EI was 12 h older than that of LI. On BA 12 days of LI (BA 12.5 days of EI), 154 eggs randomly chosen from both EI and LI were distributed into the third incubator and defined as the manipulated incubation group (MI). The remaining eggs in EI and LI were regarded as control groups. The eggs in MI were distributed with EI rows adjacent to LI rows. On BA 18 of LI, the MI group was separated into EMI group (early incubated eggs in MI) and LMI group (late incubated eggs in MI) based on the previous marks. The separation was for distinguishing the origin groups of newly hatched chicks in the hatching stage. The numbers of eggs in each group were 186 (EI), 186 (LI), 154 (EMI) and 154 (LMI).

Incubation

The eggs were incubated and hatched in lab-scale incubators (NK-hatching; Dezhou Nongke Incubation Equipment Co. Ltd., Shandong, China) measuring 1,100 × 1,000 × 900 mm with a capacity of 352 eggs. The incubators were calibrated by a standard thermometer and hygrometer before egg incubation. The incubation was maintained at a temperature of 37.8 ± 0.1 °C and a relative humidity around 60%. The turning time interval during incubation was 2 h until day 18. Eggs were candled (Cool-Lite tester; GQF, Savannah, GA, USA) on day 18 and those with a living embryo were transferred to hatching baskets. Fisheye cameras (DS-2CD3942F-I; HIKVISION, Hangzhou, China) focused upon the hatching baskets were used to monitor the hatch process. All incubations stopped at BA 504 h of LI and the chicks were removed from the incubators.

Post-hatch housing and management

A total of 128 newly hatched chicks (32 per incubation group) were sampled and transferred to four pens of 1 m2 covered with sawdust. Artificial lightning was set for 23 h/day from day 0 to 7 (40 lux at chick’s eye level). Temperature was set to 34 °C, decreased by 0.5 °C per day over seven days. Feed and water were provided ad libitum.

Data collection

On BA 18 and 20 days of LI, six eggs or chicks that hatched at peak hatching period (30–70% hatch) were randomly sampled from each group for measurements of chick embryonic development. After eggs were broken open, the embryos or chicks were sacrificed by decapitation to obtain yolk weight and yolk free body weight (YFBW). Weights of heart, liver and stomach (gizzard and proventriculus) of all sampled embryos sacrificed on BA 18 and 20 days of LI were determined.

The hatching time of individual eggs was determined using video recordings, and the hatching time was presented as BA. From the first hatchling, number of chicks was recorded every an hour. The hatched chicks were removed from incubators every 12 h to allow the camera to maintain a clear field of view. Hatch window was calculated by subtracting the hatching time of the last chick from that of the first chick. The peak hatching period was defined as 30% to 70% hatch of the batch. The total mortality was the number of dead embryos determined by candling on day 18 and unhatched eggs divided by the number of the fertile eggs.

At BA 504 h of LI (516 h of EI), 32 chicks per group which hatched in the peak hatching period were sampled and weighed. Metatarsus length (ML) was measured for assessment of leg bone development. After seven days’ growth, all chickens received the same measurements to evaluate post-hatch development.

Statistical analysis

A one-way ANOVA model (SPSS 19.0) was used to analyze the effects of egg redistribution on the embryonic development of chicks (YFBW, yolk weight, heart weight, liver weight and stomach weight) and post hatch performance (chick weight and tibia length). The level of significance was set at P < 0.05. The Fisher’s LSD method was performed to test for overall differences among treatment groups. All data are shown as average ± S. E. M.

Results

Hatch performance

The distribution of hatching time was obtained by video recordings of the four treatment groups. The EI group was found to give the first hatchling as expected, and the hatch window was 38 h (Fig. 1A). However, the hatching process of EMI group started 5 h later than that of EI group, while it finished at the same time as EI group (Fig. 1B). The start-up time of egg incubation in LI and LMI groups were 12 h later than those of EI and EMI. As a result, the first chicks of LI and LMI groups emerged from eggs 2 and 8 h later than EI group, respectively. The hatch process of LI group lasted 30 h (Fig. 1C), 8 h shorter compared to EI group. Moreover, LMI group had a shortened hatch window of 21 h with highest hatchability (95.8%), even though it started hatching at 468 h (Fig. 1D) which was 6 h later than LI. According to 30% and 70% hatch time in Fig. 2, the peak hatching period of manipulated incubation groups (EMI: 472.3–478.8 h; LMI: 475.0–480.4 h) was delayed 1.9–2.7 h compared to the control groups (EI: 470.4–477.0; LI: 472.1–478.7). Furthermore, the duration of the peak hatching period in EMI was the shortest (5.4 h) and it was consistent to the narrow hatch window (21 h).

Figure 1 Hatching pattern of chicks in four groups, including distribution of hatching time (BA), hatch window and hatchability.

(A) Early incubation group (EI); (B) early incubated eggs in manipulated group (EMI); (C) late incubation group (LI); (D) late incubated eggs in manipulated group (LMI).

Figure 2 Hatch accumulation of four groups and peak hatching period which defined as 30–70% hatch.

The red line (EMI) indicates the latest onset of the hatching process with the shortest peak hatching period.

Embryonic development from day 18 until hatch

Embryonic development of the four groups on BA 18 days of LI was shown in Table 1. YFBWs was higher in early incubation groups (EI and EMI) than those of late incubation groups (LI and LMI), but the yolk weights of early incubated eggs (EI and EMI) was found significantly lower than those of late incubated eggs (LI and LMI). In addition, organ size (heart weight and liver weight) was larger in EI and EMI, mainly caused by higher YFBW. However, no significant difference for stomach weight was found.

Table 1 Embryonic development, yolk absorption and organ weight of chick embryos on BA 18 days of LI.

	Early incubated eggs	Late incubated eggs	P value	
Control	Manipulated	Control	Manipulated	
YFBW (g)	30.27 ± 0.63a	30.50 ± 0.57a	27.68 ± 0.74b	28.41 ± 0.79b	<0.05	
Yolk weight (g)	15.16 ± 0.59b	15.46 ± 0.51b	16.31 ± 0.45a	16.69 ± 0.33a	<0.05	
Heart weight (g)	0.16 ± 0.01a	0.16 ± 0.01a	0.14 ± 0.01b	0.15 ± 0.01b	<0.05	
Liver weight (g)	0.47 ± 0.02ab	0.51 ± 0.02a	0.44 ± 0.04bc	0.42 ± 0.01c	<0.05	
Stomach weight (g)	0.95 ± 0.06	0.95 ± 0.15	0.92 ± 0.03	0.91 ± 0.11	>0.05	
Note:

a, b, c means within a row followed by different superscripts are significantly different (P > 0:05).

The chicks of four incubation groups hatched in the peak hatching period had similar YFBW (Table 2). Due to the earlier peak hatching period of chicks, yolk absorption of EI was faster and these chicks had higher liver and stomach weight. The LMI chicks that had short holding time in the incubator hatched with significantly higher yolk weight, lower liver and stomach weight. However, heart development of all hatched chicks was similar.

Table 2 Embryonic development, yolk absorption and organ weight of hatched chicks on BA 20 days of LI.

	Early incubated eggs	Late incubated eggs	P value	
Control	Manipulated	Control	Manipulated	
YFBW (g)	31.99 ± 0.89	31.38 ± 0.92	31.50 ± 0.65	31.94 ± 0.53	>0.05	
Yolk weight (g)	4.04 ± 0.26c	4.65 ± 0.20ab	4.45 ± 0.23b	4.89 ± 0.10a	<0.05	
Heart weight (g)	0.27 ± 0.01	0.27 ± 0.01	0.27 ± 0.01	0.26 ± 0.01	>0.05	
Liver weight (g)	0.93 ± 0.02a	0.88 ± 0.04ab	0.85 ± 0.04b	0.79 ± 0.02c	<0.05	
Stomach weight (g)	4.00 ± 0.05a	3.86 ± 0.26ab	3.81 ± 0.13b	3.35 ± 0.11c	<0.05	
Note:

a, b, c means within a row followed by different superscripts are significantly different (P < 0:05).

Overall, there were no significant differences between EI and EMI or LI and LMI in YFBW, yolk absorption and organ size. No effects of egg redistribution were observed for embryonic development both on BA 18 and 20 days of LI.

Post-hatch performance until day 7

The evaluation of post-hatch performance until day 7 is presented in Table 3. At peak hatching time of LMI (480 h), body weight of chicks in early incubation groups (EI and EMI) was lower due to weight loss during the holding period in hatchers, while the EI and EMI chicks had higher ML. However, no significant difference was found between EI and EMI, as well as between LI and LMI. Similar results occurred after seven days’ growth. Although both body weight and ML of early incubation groups (EI and EMI) were slightly higher than those of late incubation groups (LI and LMI), post-hatch growth and leg bone development was not altered by the manipulation of egg redistribution.

Table 3 Post-hatch body growth and leg bone development up to day 7.

	Early incubated eggs	Late incubated eggs	P value	
	Control	Manipulated	Control	Manipulated	
Day 0	
Body weight (g)	36.80 ± 0.77b	37.24 ± 0.65b	38.43 ± 0.77a	38.73 ± 0.61a	<0.05	
ML (mm)	21.64 ± 0.44a	21.53 ± 0.35a	20.85 ± 0.40b	20.82 ± 0.34b	<0.05	
Day 7	
Body weight (g)	72.66 ± 1.89	72.74 ± 1.76	72.13 ± 1.61	71.83 ± 1.25	>0.05	
ML (mm)	31.39 ± 0.45a	31.70 ± 0.57a	30.93 ± 0.57b	30.78 ± 0.63b	<0.05	
Note:

a, b means within a row followed by different superscripts are significantly different (P < 0:05).

Discussion

The aim of this study was to investigate the effects of egg redistribution during incubation on hatching time and post-hatch development. The results demonstrate that mixing eggs of different developmental stages during incubation influenced the hatching process, including delayed hatching time and shortened hatch window. They also suggest that embryonic development and post-hatch performance were not altered by the egg redistribution on BA 12 days of LI.

The hatching time is known to be influenced by factors such as parental age, egg storage time and conditions, and incubation conditions (Careghi et al., 2005; Decuypere & Bruggeman, 2007; Tona et al., 2003). The hatching time distribution also results in different chick qualities and physiological characteristics of one batch of hatched chicks (Careghi et al., 2005; Wang et al., 2014). To eliminate these factors, the eggs were obtained from a single breeder flock, laid on the same day, stored with very short time (no more than two days), and incubated in incubators with temperature and relative humidity calibration. Thus, the manipulation of egg redistribution on day 12 was presumed to be the only factor that affects the hatching time in this study.

The present study confirmed that mixing eggs of different growth curves shortened the hatch window of the redistributed group, which is consistent with hatching synchronization found in pheasants (Persson & Andersson, 1999). The onset of the hatching process of redistributed eggs was retarded 5–6 h, indicating that the narrow hatch window was related to the delay of the first hatch in manipulated group. This might be explained by some kind of egg communication between early incubated eggs and late incubated eggs. Chick embryos begin to develop a functionary auditory system as early as on day 10 of incubation (Alladi, Wadhwa & Singh, 2002). Specific interaction among the redistributed eggs may take place after mixing eggs, by means of embryo sound communication. Perception of vocalizations by embryos may lead to physiological or behavioral changes. This is consistent with the finding of Tong et al. (2015a) that internal piping time was delayed when embryos were exposed to manmade sound stimulation of embryos. However, increased mortality was observed in duck and chicken eggs that were incubated under artificial sound stimulation (Tong et al., 2015a; Veterany, Hluchý & Weis, 1999). Compared to the artificial sound stimulation, embryo vocalization may impose less stress on other hatching eggs and exerts no negative impact on hatchability. Another hypothesis is that environmental CO2 alters the hatch process and results in a narrow spread of hatch. Previous researchers reported that high levels of CO2 during the early stages of incubation stimulated early hatching and shortened hatch window (De Smit et al., 2006; Tona et al., 2007). Although the onset of hatching process of mixed eggs was delayed compared to the control groups, this did not extend the spread of hatch. The early incubated embryos may penetrate the membrane and eggshell, and generate more CO2 during the hatching period, leading to increased CO2 concentration that stimulated the hatching process of late incubated eggs. Furthermore, increasing the CO2 concentration potentially contributes to the hatchability of LMI (95.8%)—higher than the other groups—suggesting that more chick embryos succeeded in breaking out of eggshell rather than died in this difficult process. Considering this delayed onset of the hatching process, the narrow spread of hatch and the increased hatchability, our future work will focus on identifying to what degree, and via which mechanisms, redistributing eggs of different growth curves affects hatching pattern and hatchability.

The advanced embryonic development of early incubated eggs was observed in both control (EI) and manipulated group (EMI), mainly caused by the initial incubation time difference of 12 h. However, mixing eggs of different growth curves did not alter the embryonic growth and yolk absorption before hatch. Chick embryos of both early incubated and late incubated eggs were able to maintain normal organ development and nutrient metabolism until hatch. Although the earlier hatched chicks (EI and EMI) underwent a longer holding period in incubators, the decreased yolk weight and increased organ weight indicated that they got advanced maturation of organs after hatching, as supposed by previous studies (Pinchasov & Noy, 1993; Tong et al., 2015b; Van de Ven et al., 2011). No access to feed and water (EI and EMI, 36 h; LI and LMI, 24 h) for a long time resulted in a higher weight loss in early incubation groups (EI and EMI), but enhanced the leg bone development. The consistency of body weight and leg bone development on day 7 was observed as expected. Nevertheless, the narrow hatch window of manipulated groups did not influence chick growth performance up to day 7, indicating that egg distribution only stimulates the hatching behavior. However, there is no evidence that response to eggs or egg communication by egg distribution was related to this shortened hatch window. As reported above, there was no negative effect of mixing eggs of different growth curves on embryonic growth, utilization of nutrients and post-hatch performance.

Conclusion

The specific manipulation of mixing eggs of different initial incubation time influenced the hatching pattern of late incubated eggs, including advanced hatching process and narrow hatch window, but did not affect normal embryonic development, utilization of nutrients and post-hatch performance of the late incubated eggs. All of these results are applicable in the industrial hatchery to shorten hatch window and improve the uniformity of chicks.

Supplemental Information

Supplemental Information 1 Raw data.

Click here for additional data file.

We would like to thank Kailao Wang for assistance of video camera. We also thank Zhanming Li (College of Quality & Safety Engineering, China Jiliang University) for his suggestions.

Additional Information and Declarations

Competing Interests

Author Contributions

Animal Ethics

Data Availability

The authors declare that they have no competing interests.

Zhentao Zhong conceived and designed the experiments, performed the experiments, analyzed the data, contributed reagents/materials/analysis tools, prepared figures and/or tables, authored or reviewed drafts of the paper, approved the final draft.

Yue Yu performed the experiments, prepared figures and/or tables, approved the final draft.

Shufang Jin performed the experiments, prepared figures and/or tables.

Jinming Pan conceived and designed the experiments, analyzed the data, contributed reagents/materials/analysis tools, authored or reviewed drafts of the paper, approved the final draft.

The following information was supplied relating to ethical approvals (i.e., approving body and any reference numbers):

All procedures in this study were approved by the committee of the Care and Use of Animals of Zhejiang University, Hangzhou, China.

The following information was supplied regarding data availability:

The raw data are provided in the Supplemental Dataset File.

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
