# Peer review of "Effects of mixing eggs of different initial incubation time on the hatching pattern, chick embryonic development and post-hatch performance"

_PeerJ, doi:10.7717/peerj.4634_

## Round 0.1 · original submission · Major Revisions

There are a number of issues with language that will have to be thoroughly addressed during revision. Description of methods also is a major issue with the current version, which caused Reviewer 1 to recommend Rejection. Please address this major points and others raised by the reviewer.

Reviewer 1 ·

Basic reporting

The quality of written English is basic, often with missing words and unconjugated verbs. This is visible throughout the whole text but particularly noticeable in lines 9, 16-18 (abstract), lines 26-36 (introduction) and lines 158 and 163 (discussion).
The methods section in the abstract chapter requires more detail.
The introduction section is not focused and does not fully explain the background supporting the work.
Information that should be in the Materials and Methods section (such as the fact that the eggs were laid on the same day and stored for no longer than 2 days) is instead featured in the Discussion, in lines 155-156.

Experimental design

The research is original and the research question is well defined.
However, the description of the methods does not allow for sufficient detail to replicate the study as the exact number of eggs in each study group is not divulged (only the number of chicks used for the post-hatching trials), although it can be estimated from raw data. Also, there is no mention of mortality rates or even how (or if) the EMI and LMI eggs were marked so that they could be later separated for hatching. There is also no explanation offered on why these two groups were separated for the hatching stage.

Validity of the findings

The data is robust and statistically sound.
However, it is not mentioned that different numbers of eggs were incubated/set to hatch for each group (as calculated from the number of eggs hatched vs hatchability per group - EI=151, EMI=126, LI=139, LMI=118) and it is not discussed how a larger number of eggs could influence the hatch window.
Discussion of the results is mostly supported by speculation around sound stimulation and CO2 stimulation of embryos. However, it does not address the fact that multi-stage incubators are still often used, mixing eggs which have started incubation 24, 48, 72 hours apart. Is this not one of the main reasons why a study like this would be meaningful?
The conclusions taken are limited and not quite clear.

·

Basic reporting

This is an interesting study that provides additional information about ways of decreasing the hatch window and increasing the uniformity of chicks in order to ensure good post-hatch performance.
The manuscript is generally well-written although I have made many minor suggestions for changes to increase the clarity and readability of the paper. Sufficient background and context are provided and the results are relevant to the stated aims.

Experimental design

The experimental design is appropriate to the stated aims of the study. The study appears to have been performed to a high technical and ethical standard.

Validity of the findings

The reported data appear to be robust and the discussion and conclusion are consistent with the findings.

Additional comments

The manuscript is generally well-written. However, for authors for whom English is not the first language, it is always helpful to have the manuscript read by a native English speaker, if at all possible.

---

## Round 0.2 · Minor Revisions

Please address the remaining issues with language, which should be carefully edited throughout

Reviewer 1 ·

Basic reporting

English still needs revising as there are still various points where words are missing or in excess. For instance, the sentence in lines 39 to 42 is not a sentence, as it lacks a verb.
Likewise, in the Materials and methods section, line 66, the EI group is wrongly labeled as the LI group.

Experimental design

The total mortality should be divided between: 1) dead-in-shell embryos and 2) unhatched eggs.

Validity of the findings

No comments

·

Basic reporting

The manuscript is improved by the authors' adopting the changes suggested by the two reviewers.

Experimental design

The experimental design is clearly stated and ambiguities identified by the reviewers have been addressed.

Validity of the findings

The reporting of the findings has been modified to clarify identified ambiguities.

Additional comments

The manuscript is now suitable for publication in PeerJ.

---

## Round 0.3 · accepted · Accept

Thanks for addressing the final minor concerns.

#